Subject Area:
cellular biology/developmental biology/
genetics

Keywords:
planar cell polarity, *Drosophila* eye, ommatidial rotation, integrins, extracellular matrix

Author for correspondence:
Marek Mlodzik
e-mail: marek.mlodzik@mssm.edu

†These authors contributed equally to this work.
‡Present address: The Swedish Research Council, Stockholm, Sweden.
¶Present address: CSL Behring, Keelung Road, Taipei City 110, Taiwan, Republic of China.
∥Present address: 3401 Leafwood Court, San Mateo, CA 94403, USA.

# Integrins are required for synchronous ommatidial rotation in the *Drosophila* eye linking planar cell polarity signalling to the extracellular matrix

Maria Thuveson[1,†,‡], Konstantin Gaengel[1,2,†], Giovanna M. Collu[1], Mei-ling Chin[1,¶], Jaskirat Singh[1,∥] and Marek Mlodzik[1]

[1]Department of Cell, Developmental and Regenerative Biology and Graduate School of Biomedical Sciences, Icahn School of Medicine at Mount Sinai, Annenberg Building 18-92, One Gustave L. Levy Place, New York, NY 10029, USA
[2]Department of Immunology, Genetics and Pathology, Uppsala University, Rudbeck Laboratory C11, Dag Hammarskjölds Väg 20, 751 85 Uppsala, Sweden

MM, 0000-0002-0628-3465

Integrins mediate the anchorage between cells and their environment, the extracellular matrix (ECM), and form transmembrane links between the ECM and the cytoskeleton, a conserved feature throughout development and morphogenesis of epithelial organs. Here, we demonstrate that integrins and components of the ECM are required during the planar cell polarity (PCP) signalling-regulated cell movement of ommatidial rotation in the *Drosophila* eye. The loss-of-function mutations of integrins or ECM components cause defects in rotation, with mutant clusters rotating asynchronously compared to wild-type clusters. Initially, mutant clusters tend to rotate faster, and at later stages they fail to be synchronous with their neighbours, leading to aberrant rotation angles and resulting in a disorganized ommatidial arrangement in adult eyes. We further demonstrate that integrin localization changes dynamically during the rotation process. Our data suggest that core Frizzled/PCP factors, acting through RhoA and Rho kinase, regulate the function/activity of integrins and that integrins thus contribute to the complex interaction network of PCP signalling, cell adhesion and cytoskeletal elements required for a precise and synchronous 90° rotation movement.

## 1. Introduction

Rotation of ommatidial preclusters is the final read-out of planar cell polarity (PCP) establishment in the *Drosophila* eye [1–5]. It is an excellent model for the study of morphogenetic movements regulated by PCP signalling. Besides well-studied border cell migration processes during *Drosophila* oogenesis [6–8], ommatidial rotation (OR) provides an interesting *in vivo* model to study complex regulatory interplays of cell adhesion and signalling pathways, leading to highly coordinated movement of groups of cell [5]. The *Drosophila* eye develops from a single-layer epithelium, the eye imaginal disc, proceeding in a highly organized spatio-temporal manner. The differentiation sequence is marked at its anterior by the morphogenetic furrow (MF). Posterior to the MF, photoreceptor precursors are progressively assembled into ommatidial preclusters and specified as individual R cells. The clusters begin to rotate in the five-cell precluster (R8, R2/R5, R3/R4), just after the Fz/PCP signalling-mediated R3/R4 specification is established and rotate as units precisely 90° from their initial position [1–3] (see also figure 1*a* for sequence of ommatidial assembly). Ommatidia on each side of the dorsal–ventral midline, the equator,

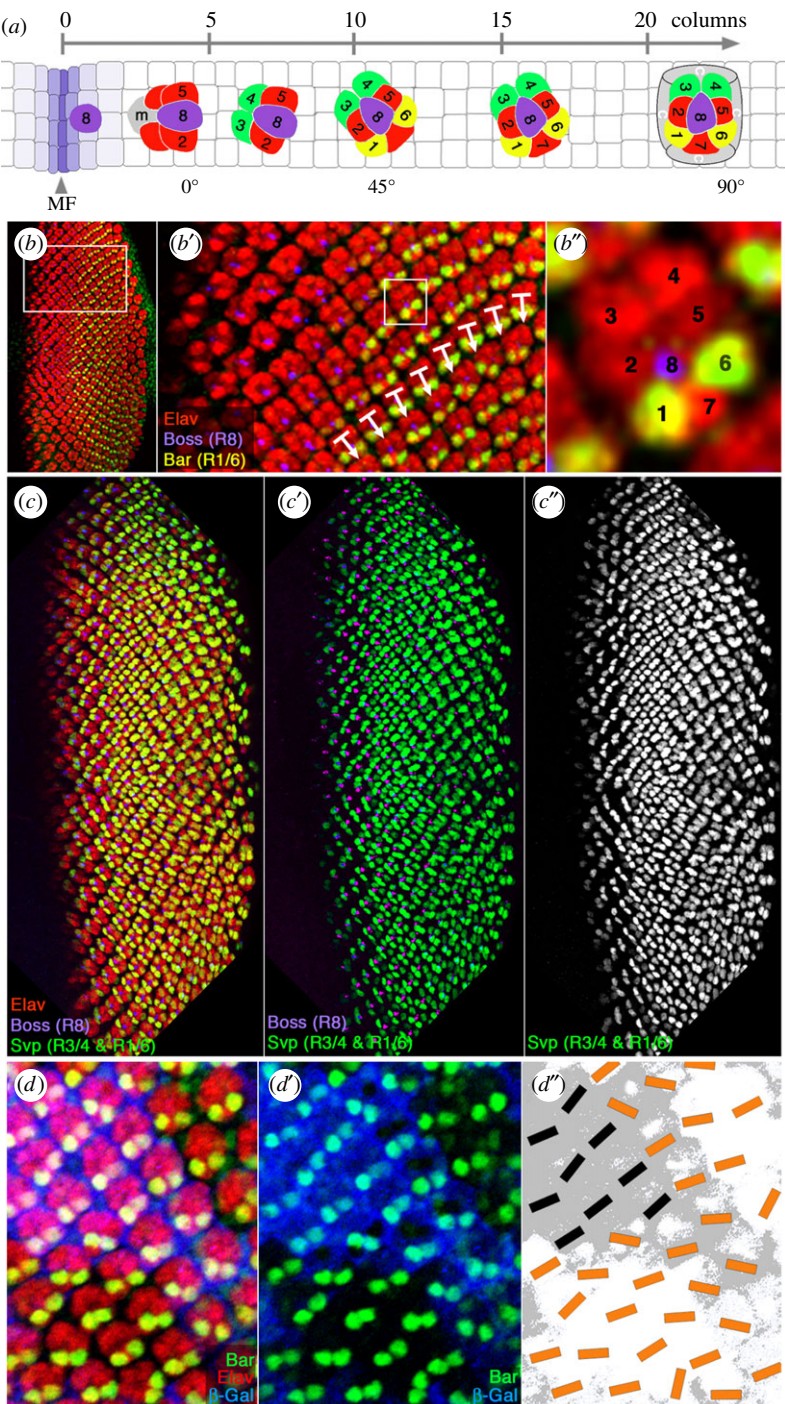

**Figure 1.** Loss of *βPS/mys* integrin causes defects in OR. (*a*) Schematic presentation of photoreceptor (R-cell) differentiation posterior to MF. Anterior is left and dorsal up in all panels. The first R-cell to be specified is R8 (shown in purple). With the induction and specification of R2/R5 and R3/R4 pairs, the five-cell precluster is established (mystery cells (m) are initially also part of the precluster but later excluded). Subsequently, the R1/R6 pair, R7 and cone cells (c) are added. Ommatidia rotate an initial 45°, usually completed by columns 9–10 and then continue rotation until they reach their final 90°. Cartoons of typical ommatidia from columns 4, 7, 11, 16 and 22 are shown in third instar eye imaginal discs. The respective photoreceptors, R cells, are numbered. (*b–b″*) Confocal microscopy image of the third instar eye imaginal disc stained with anti-Elav (red: marking all R cells), anti-Bar (green; staining R1/R6 cells) and anti-Boss (purple; central R8 cell). (*b′*) Enlargement of boxed area in *b*; white arrows indicate rotation angles. (*b″*) Single ommatidium (boxed in *b′*). Numbers indicate respective R cells. (*c–c″*) Confocal microscopy image of wild-type third instar eye imaginal disc stained with anti-Elav (red: marking all R cells), *svp-lacZ* (green; staining the R3/R4 and R1/R6 cell pairs) and anti-Boss (purple; central R8 cell). (*c′*) The same eye disc, but lacking the Elav staining to easier appreciate OR angles, and (*c″*) is a monochrome of *svp-lacZ* staining. Note that both, anti-Bar (*b–b″*) and *svp-lacZ* (*c–c″*) are excellent markers to measure the respective rotation angles of individual ommatidia. See also electronic supplementary material, figure S1 for additional marker combinations used. (*d–d″*) Higher magnification area of eye imaginal disc with *mys/βPS* mutant clones (*mys¹* allele is a reported protein null). Mutant tissue is marked by lack of β-gal staining (blue in *d,d′* and shaded in grey in schematic in *d″*). R cells are marked with anti-Elav (red in *d*), and anti-Bar (green in *d,d′*, staining the R1/R6 pair and reflecting the rotation angle). (*d″*) Schematic of the rotation angles in wild-type (grey shaded area) versus mutant (white area) with bars reflecting angle; orange bars indicate rotation angles of R1/R6 pairs in *mys¹* mutant or mosaic clusters; black bars mark orientation in fully *wt* clusters. Note that whereas black bars (wild-type) are close to a 45° at this developmental stage, orange bars display an abnormally wide range of angles (often reflecting an over-rotation of an ommatidium). Also, note that the majority of ommatidia in *mys/βPS* integrin mutant tissue are out of synchrony when compared with the neighbouring clusters. While some reach the final 90° rotation angle earlier than ommatidia of equivalent developmental stages in adjacent wild-type tissue, others lag behind (see also figures 2 and 3 for quantifications).

rotate in opposite directions resulting in a mirror image arrangement across the equator, with direction of rotation being determined by the cell fate within the R3/R4 pair [1–3]. The five-cell preclusters are initially bilaterally symmetric and the pair that will become R3/R4 are equivalent, before one of the two cells of the R3/R4 pair, the one that is closer to the equator, adopts the R3 fate (via Frizzled/PCP signalling), while its neighbour will be specified as R4 via Notch signalling [9–13].

Although PCP signalling-dependent R3/R4 cell fate determination is relatively well understood [4,14,15], the downstream event of OR remains poorly understood mechanistically. Only a few 'rotation-specific' genes have been identified that affect OR, but not R3/R4 specification. These include the core PCP signalling effector *nemo* (*nmo*; Nlk in vertebrates), a Ser/Thr kinase related to MAPK [16–18] and E-cadherin [19], which both promote rotation. In addition, Egfr/Ras signalling is a key regulator of OR [20–22], although with a less well-defined role. However, both the PCP effector Nmo and Egfr signalling likely control rotation (at least in part) through cadherin-based adhesion, as suggested by functional studies with both E-cadherin and the atypical cadherin Flamingo [18,20,21]. Rho kinase (*dRok*) serves as a potential link between Fz/PCP signalling and the cytoskeleton [23]. dRok functions in a subset of PCP events: for example, it regulates the degree of rotation, but not the R3/R4 cell fate specification, and similarly it controls wing hair number, but not hair orientation [23], possibly via Myosin-II regulation [24]. Rho family GTPases are known to act by remodelling the actin cytoskeleton during cell migration (e.g. reviewed in [25]).

Integrins contribute to many morphogenetic events that shape a developing organism and its organs [26–33]. Besides cell–cell adhesion being critical for morphogenetic events (e.g. via cadherins), the anchorage between cells and their extracellular environment is also essential for morphogenesis. Integrins act by binding cells to the extracellular matrix (ECM), a highly ordered structure of proteins and glycosaminoglycans that form an insoluble meshwork in the extracellular space between cell layers or groups of cells [27,29–31]. Integrins form the transmembrane link between the ECM and the actin cytoskeleton of the cell. Changes in cell–cell and cell–matrix contacts generally affect cell movements, cell shape and cell signalling, and consequently the alignment and integrity of tissues; because integrins mediate cell–ECM contacts, losing integrin function results in various developmental defects [29–33]. Integrins do more than simply forming a mechanistic link between the ECM to the actin cytoskeleton. They also convey signals, directly through associated intracellular kinases and indirectly by promoting adhesion to the ECM so that cells are kept close to the source of signals. The binding of ECM–ligands to integrins leads to integrin clustering and recruitment of intracellular proteins to the short cytoplasmic domain of the β subunit of the integrin heterodimer [28]. Integrins 'transduce' signals by spatially compartmentalizing docking and adapter proteins that link integrins to cytoplasmic kinases, GTPases and other enzymes [34,35]. In addition to providing anchorage necessary for cell migration, integrin engagement activates pro-migratory signals regulating cytoskeletal elements and cell movements.

Molecularly, integrins function as non-covalently linked transmembrane proteins, composed of α- and β-subunits.

α/β-heterodimers are assembled in the endoplasmic reticulum and only as heterodimers are they transported to the cell surface [36]. In *Drosophila*, two β-subunits (βPS and βν) and five α-subunits (αPS1–5) have been identified. The major β-subunit in the fly is βPS, the orthologue of β1 in vertebrates, which is widely expressed throughout development and can form dimers with all five α-subunits (reviewed in [29,37]). The βν-subunit is required in the midgut, where it pairs with αPS3 [30,38,39] and in hematopoietic cells [40]. The absence of the βPS subunit results in detachment and rounding up of muscles, among other defects; hence, the gene encoding βPS is called *myospheroid* (*mys*) [41,42]. Overall, studies in *Drosophila* muscle attachment and maturation have provided seminal insights into the role of integrin function in development and organogenesis (e.g. reviewed in [32,33]).

Many processes require integrins in *Drosophila*, including adhesion of the dorsal and ventral wing-layers (with integrin mutants causing wing blisters), muscle tendon interactions or integrin-dependent anchoring to the stem cell niche in the germ line (e.g. [26,27,30,32,33,43–46]). Integrins are also critical for various morphogenetic processes such as dorsal closure, retraction of the germ band or migration of primordial midgut cells [39,47–49].

Cell motility is driven by remodelling of the cytoskeleton and often requires contacts with the ECM [50]. Moreover, Frizzled/PCP signalling has been suggested to cooperate with ECM components during the processes of convergence and extension during gastrulation and neurulation [51,52]. We thus tested for potential roles of integrins in the process of OR and their potential interplay with PCP signalling.

Here, we demonstrate that integrins and the link they form between the ECM and the cytoskeleton are required in this developmentally regulated cell movement of OR. The loss-of-function integrin ommatidial clusters, as evident in clones of a *mys*/βPS null allele, rotate much more irregularly than wild-type controls with a wide spread of rotation angles at each stage. In addition, mutant clusters tend initially to rotate faster than their wild-type neighbours. These data suggest that integrins contribute to a tight temporal regulation of rotation. The localization of integrins changes dynamically during OR and they form a basolateral goblet-like structure surrounding the outside of each precluster. Our data suggest that the function of integrins is linked to PCP–RhoA–dRok signalling and their interaction with the ECM is required for the temporal regulation of rotation and thus the spatio-temporal precision and synchrony between clusters.

# 2. Material and methods

## 2.1. *Drosophila* husbandry, stocks and genetics

The following stocks were used: *UAS-mys* and *UAS-torsoᴰβPS_cyt* (gifts from F. 1991 Schoeck) [49], *UAS-RhoAᴵᴿ* and *UAS-dRokᶜᵃᵗ* (gifts from L. Luo) [53], *tigᴬ¹* (gift from T. Bunch), *wb⁴ʸ¹⁸* (gift from F. Schoeck), *mys¹*, *mysⁿᵘˡˡ*, *mewᴹ⁶*, *if³*, *if²*, *vinc¹*, *stck³ᴿ⁻¹⁷*, *ILK¹*, *trol¹³*, *wb⁰⁹⁴³⁷*, *sdc¹⁰⁶⁰⁶*, *fzᴾ²¹*, *fzᴿ⁵²*, *stbmⁿᵘˡˡ*, *fmiᴱ⁴⁵*, *RhoA⁷²⁰*, *dRok²*, *zip¹*, *sqhᴾᴸ⁹¹*, *chic⁰¹³²⁰*, *svp⁷⁸⁴²*, *actin > CD2 > Gal4; UAS-GFP, hs-FLP, sev-Gal4, ey-Flp*. If not otherwise noted, all other fly stocks are as described in Flybase (http://flybase.bio.indiana.edu/),

and were provided by the Bloomington Stock Center (http://flystocks.bio.indiana.edu). Crosses for genetic interactions were performed at 29°C unless otherwise indicated; LOF and GOF clones were generated at 25°C.

Mutant eye clones were generated with the *ey-FLP* technique [54]. The following chromosomes were used:

FRT19A, *mys¹* (strong LOF allele) of βPS–integrin.
FRT40A, *wb⁴ʸ¹⁸* (strong LOF allele).
GOF clones of *UAS-dRok^{cat}* were generated with the 'Flip-out' technique [55] using an *hs-FLP, actin > CD2 > Gal4: UAS-GFP* strain.

The *sevenless(sev)-Gal 4* driver was used to drive the expression of *UAS-mys, UAS-torso^D βPS_{cyt}* and *UAS-RhoA^{IR}*. *sev-Gal4* is expressed strongly in R3/R4, R7 and cone cells, and weaker in R1/R6 [56].

## 2.2. Immunohistochemistry, histological and statistical analyses

The dissection of imaginal discs and antibody stainings were performed as described [21] and mounted in Vectashield (Vector Labs). Secondary antibodies coupled to FITC, TRITC and Cy5 were from Jackson Laboratories. Primary antibodies were: mouse anti-βPS/mys (1 : 200), rat anti-DE-cadherin (1 : 20), mouse anti-FasIII (1 : 100); rat anti-Elav (1 : 200), mouse anti-Boss (1 : 1000), mouse anti-Fmi (1 : 10; all from DSHB), rabbit anti-Bar (gift from K. Saigo; 1 : 100), mouse or rabbit anti-βGal (Promega and Molecular Probes; 1 : 2000) and rabbit anti-GFP (Molecular Probes; 1 : 4000). Images were acquired with a Zeiss LSM510 confocal microscope and assembled in Adobe Photoshop.

The numerical values of OR angles in individual clusters were determined with the 'angle' option in Photoshop or ImageJ. For each genotype and molecular marker, several eye discs with many clusters were analysed and plotted. Statistical analyses were performed using the Student *t*-test for averages, and the Kolmogorov–Smirnov test for distribution and spread of angle values.

For adult eye sections, the tissue embedding and sectioning of eyes was performed as described [57]. Multiple eyes from female flies were sectioned and analysed for each genotype with *n* values higher than 100 in each case.

## 3. Results

### 3.1. Loss of *mys*/βPS–integrin causes defects in ommatidial rotation

The rotation of ommatidial preclusters is initiated in the third instar eye imaginal disc shortly after the specification of the R3/R4 pair (schematic in figure 1*a*). It starts with a fairly fast initial rotation in the fifth column, which is then followed with a slower rotation phase when ommatidia progressively continue their rotation until reaching their final 90° position from its initiation (scheme in figure 1*a*). In developing eye discs, the rotation angles (or orientation) of preclusters with respect to the equator can be determined using specific markers for the R3/R4 and R1/R6 pairs. Examples of markers include anti-Bar (highlighting R1/R6 pairs from column 10 onward; figure 1*b–b″*) [58] or with *svp-lacZ* (highlighting

R3/R4 early from column 5 onward and also R1/R6 starting in column 10; figure 1*c–c″*; electronic supplementary material, figure S1C) [59]. Additional useful markers that highlight the orientation and structure of the preclusters include the trans-membrane or membrane-associated proteins E-cadherin and Canoe::GFP (electronic supplementary material, figure S1A,B). The staining of any of these marker combinations, with Elav marking all R cells, for example, allows precise measurements of the rotation angles of individual clusters and associated quantifications (see also electronic supplementary material, figure S1 displaying rotation angle measurements).

Integrins are required for the morphogenesis of photo-receptors. They attach the cellular base of photoreceptors and their rhabdomeres to the retinal floor. This cell–ECM contact is critical for elongation of the rhabdomeres, which accompanies retinal deepening during late retinal morphogenetic processes [60,61]. For this reason, adult eye sections of integrin mutant clones have a very disorganized appearance and thus cannot be examined for OR-associated patterning. Thus, to investigate a potential role of integrins in OR, we turned to clonal loss-of-function studies in larval eye discs during the active OR process, using the marker combinations described above. Since only αβ-heterodimers are transported from the endoplasmic reticulum to the cell surface, inducing loss-of-function clones using a βPS/mys null allele completely abolished integrin function [39]. Homozygous *mys¹* clones are of similar size as their wild-type (*wt*) controls, indicating no major defects in proliferation or cell survival, and, importantly, epithelial integrity was also normal at this stage (see below). Our analyses of eye imaginal disc clones of a βPS/mys null allele (*mys¹*) revealed rotation defects (figure 1*d,d′*). Strikingly, ommatidial clusters in βPS/mys mutant tissue rotated less synchronously than in *wt* tissue, with clusters rotating both more quickly and more slowly (figure 1*d–d″*, with detailed quantifications in figures 2 and 3).

Large-scale quantitative analyses of βPS/mys clones revealed two main defects: (i) in early columns, columns 5–9, mutant clusters appeared to rotate faster than their *wt* control clusters (figure 2; electronic supplementary material, figures S2A,A′ and S3) and (ii) mutant clusters displayed a significantly wider range and distribution of rotation angles at any given stage of the process, which becomes more exacerbated from column 10 onward (figure 3; electronic supplementary material, figures S2 and S3), indicating that the temporal (synchronous) control of OR is impaired (compare angle distribution and spread in quantifications in figures 2 and 3; electronic supplementary material, figure S3). For example, while some βPS/mys mutant ommatidial clusters could reach the final 90° as early as column 11, whereas *wt* control clusters displayed angles of 40–60° at this stage (figures 2 and 3), other mutant clusters lagged behind their *wt* counterparts (figure 3; electronic supplementary material, figure S3). The rotation angle distribution and spread were significantly increased in both fully mutant or mosaic clusters (including mosaic clusters where only few R cells were mutant; e.g. figure 1*d–d″*, coloured bars over grey shaded areas, when compared with their *wt* counterparts).

Taken together, these analyses established that OR was defective in βPS/mys mutant tissue, with the rate and synchrony of rotation being affected throughout the process. This suggested that cell–ECM interactions, mediated by the integrin complexes, are critical for this cell motility process.

royalsocietypublishing.org/journal/rsob   Open Biol. 9: 190148

royalsocietypublishing.org/journal/rsob   Open Biol. **9**: 190148

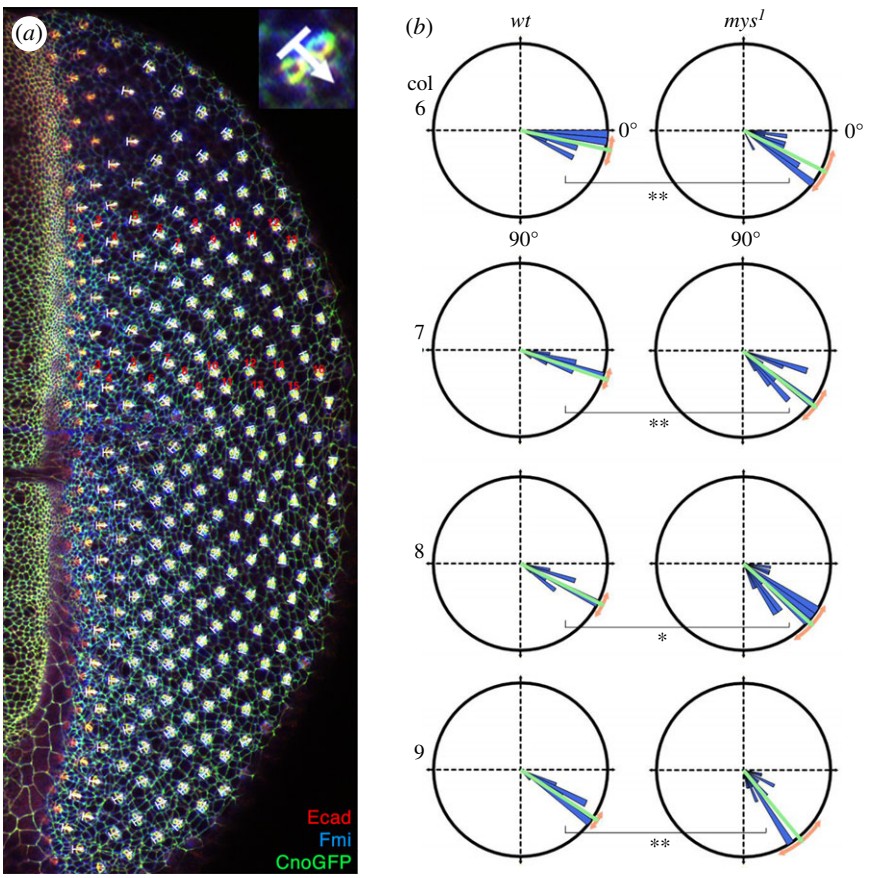

**Figure 2.** βPS/mys integrin mutant clusters tend to rotate too far at early stages. (a) Confocal microscopy image of wild-type third instar eye imaginal disc stained with anti-E-cad (red), anti-Fmi (blue) and CnoGFO (green), representing typical wild-type examples of our quantifications with rotation angles (white arrows) measured in each cluster in the respective columns (marked by red numbers within two regions of the dorsal eye area: mid dorsal and more central, respectively). Inset in top right corner delineates how angle is determined relative to the E-cad/CnoGFP staining, which strongly marks the adherens junctions of R2/R5. For quantitative analyses, see also electronic supplementary material, figure S1, which shows the same disc with the respective rotation angle values (electronic supplementary material, figure S1A), and rotation angle arrows and values for discs stained for ant-Elav co-stained with anti-Bar (R1/R6 marker) and CnoGFP (electronic supplementary material, figure S1B) and svp-lacZ, anti-Elav and anti-Boss (electronic supplementary material, figure S1C). Equivalently, marked discs containing mutant *mys*[1] clones were used to establish rotation angles in *mys*/βPS mutant tissue (examples shown in electronic supplementary material, figure S2). (b) Comparative quantification of rotation angles of preclusters in columns 5–10 in *wt* and *mys*[1] mutant tissue presented in rosette diagrams. The presented intervals are in 5° wedges; the average is shown by green line and standard deviation (as a measure of angle spread and variation) in orange. 0° is horizontal right (as in MF at onset of rotation) and 90° (as in adults after completion of OR) is down in all examples. Columns 6–9 of *wt* and *mys*[1] mutant clusters are shown. Note that clusters in *mys*[1] mutants displayed generally a wider spread of rotation angles and at these early stages had a tendency to rotate farther early on. Statistical analyses with Student's *t*-test, *p*-values were: *<0.011 and **<0.001. See also electronic supplementary material, figure S3 for additional samples of comparative rotation angle quantifications.

## 3.2. Expression of dominant-negative βPS/*mys* affects the rotation process

Since adult eye sections of βPS/*mys* loss-of-function mutant tissue cannot be analysed for rotation defects due to severe malformations during rhabdomere morphogenesis, we turned to a misexpression approach to further corroborate the role(s) of integrins in OR. The overexpression of a wt *UAS-mys* transgene acts in a dominant-negative manner [62]. For example, wing blisters were observed when βPS/ mys was overexpressed in developing wing tissue [62], consistent with a mild loss-of-function phenotype of integrins.

Thus, to further investigate integrin function in OR, we examined flies that expressed a UAS-*mys* transgene under control of the ommatidial precluster-specific *sevenless-Gal4* driver (*sev > mys*). Although a single copy of *sev > mys* resulted in minimal eye defects at 25°C (figure 4a), two copies of the *UAS-mys* transgene caused robust rotation

defects, with approximately 50% of ommatidia being misrotated. Both under- and overrotated ommatidia were observed (figure 4b and table 1; quantified in figure 3b), consistent with our observation of asynchronous and randomized, wide distribution of rotation angles in eye imaginal discs (see above). Rotation defects, albeit weaker than two copies at 25°C, were also observed with a single copy of *sev > mys* at 29°C (figure 4c). The overexpression phenotype was enhanced in a *mys* heterozygous background, confirming that it acts as a dominant negative, and this was further supported by the enhancement of the *sev > mys* phenotype by a removal of one copy of either of the α subunits, αPS1 or αPS2 (encoded by the *mew* and *if* genes, respectively; figure 4d and table 1). In addition to the strong rotation defects seen with *sev > mys* (2 copies of *UAS-mys* at 25°C), there was also loss of photoreceptors observed in these eyes: approximately 27% of ommatidia were missing one or more photoreceptor cells, often R7 (see examples in figure 4b,d), suggesting that integrin function

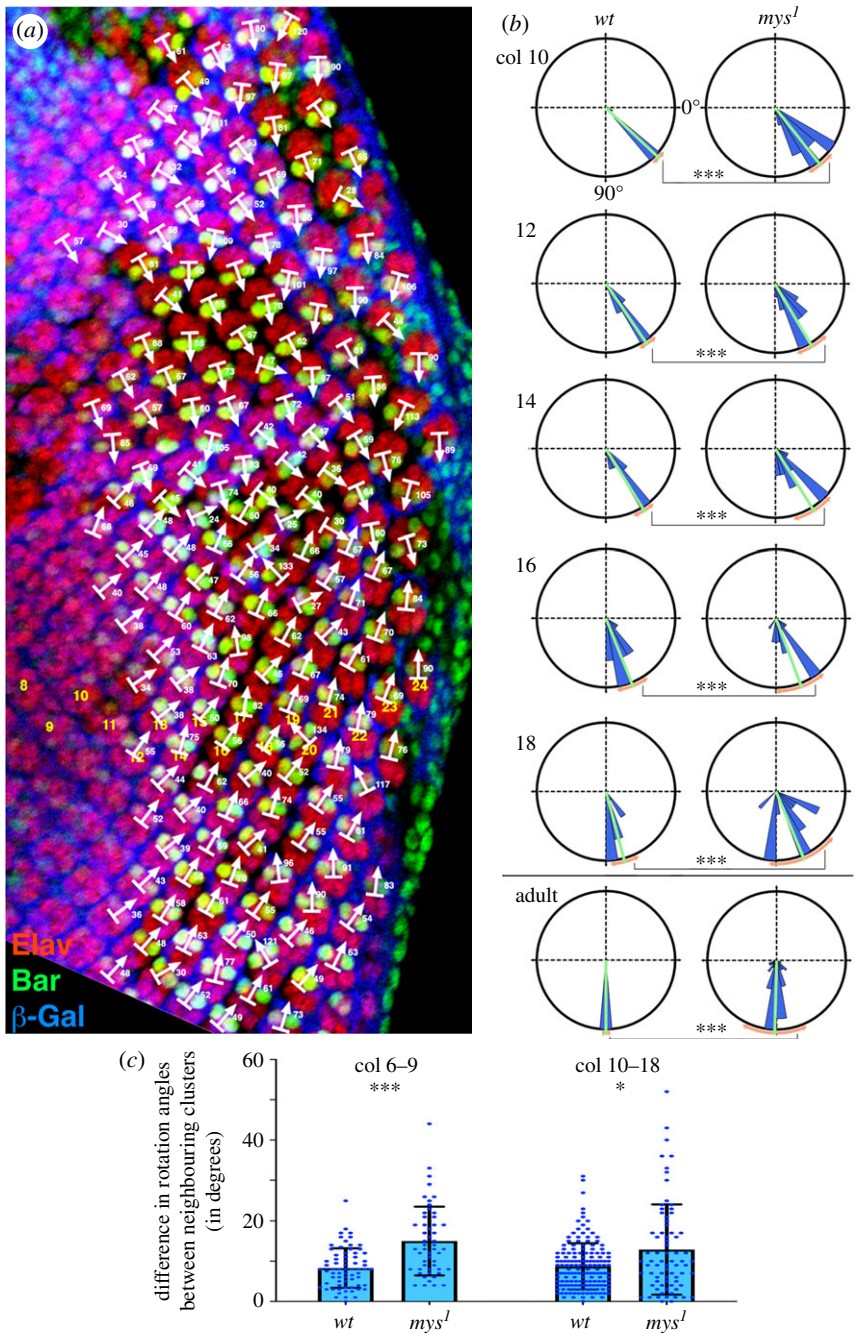

**Figure 3.** βPS/*mys* integrin mutant clusters do not rotate synchronously. (*a*) Confocal microscopy image of the third instar eye imaginal disc containing *mys¹* mutant clones (marked by the absence of β-Gal expression, blue) stained with anti-Elav (red: marking all R cells) and anti-Bar (green; staining R1/R6 cells). The disc represents a typical example of mutant tissue analyses with rotation angles measured in each cluster (white arrows, angle values indicated), the respective columns are numbered in yellow in ventral disc area. See also electronic supplementary material, figure S2 for additional examples of discs containing *mys¹* mutant clones. (*b*) Comparative quantification of rotation angles, presented as rosette diagrams, of clusters in columns 10, 12, 14, 16 and 18, and adult eyes in *wt* and *mys¹* mutant tissue (the adult phenotype was recorded in *sev-mys* flies, which are dominant negative for βPS/Mys function, as analysed in figure 4). The presented intervals are in 10° wedges (blue); the average is shown by green line and the standard deviation (measure of angle spread and variation) in orange. 0° is horizontal right (representing angle prior to OR, see also figure 2) and 90° (representing completion of OR) is down in all samples. Note the much wider rotation angle distribution in *mys¹* tissue (or in *sev-mys* adult eyes) when compared with the equivalent *wt* stages. See electronic supplementary material, figure S3 for additional examples of comparative rotation angle quantifications. (*c*) Comparative quantification of the difference in rotation angles between a cluster and its nearest neighbours within the same column. Data are grouped into columns 6–9 (left) and 10–18 (right). Individual data points are plotted in dark blue and the mean with standard deviation are shown ($n = 69$ and 158 for *wt* and $n = 52$ and 74 for *mys¹*). Statistical studies were performed with the Kolmogorov–Smirnov analyses, with focus on angle variation distribution. *p*-values were: *<0.05 and ***<0.0001. Note that average rotation angles were largely not different between *wt* and mutant, but the angle distribution was significantly changed, and there was greater variation in angle compared to the neighbouring clusters.

was possibly required for proper positioning of the R7 precursor cell and thus recruitment of R7 (also below).

Taken together with the eye disc loss-of-function *mys¹* clone phenotypes, these data confirm that integrins are required for precise and synchronous rotation, as the angle distributions in mutant conditions are much wider at all stages analysed, including the adults (compare quantifications in figure 3; also electronic supplementary material, figure S3).

royalsocietypublishing.org/journal/rsob    Open Biol. 9: 190148

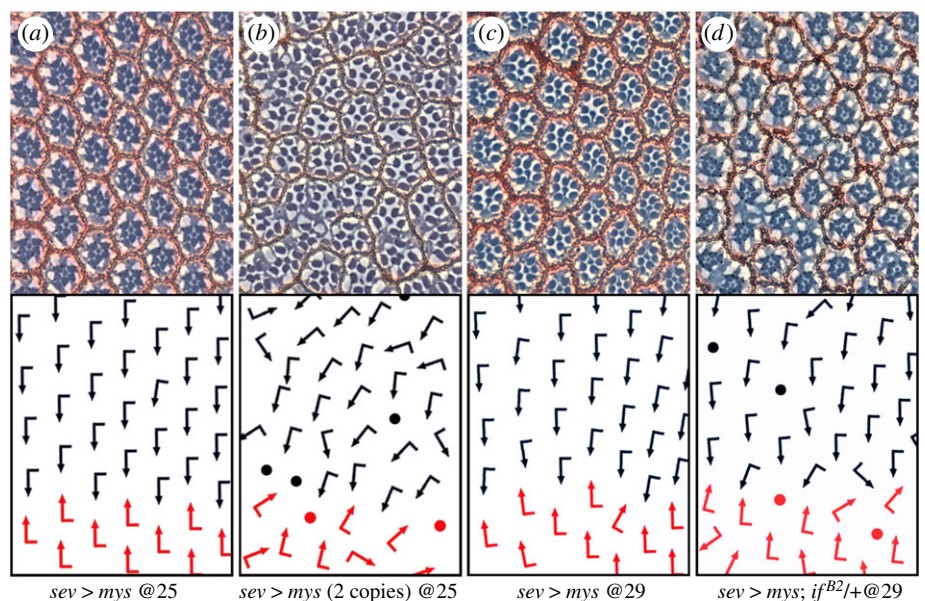

| (a) | (b) | (c) | (d) |

*sev > mys* @25        *sev > mys* (2 copies) @25        *sev > mys* @29        *sev > mys*; *if^{B2}*/+@29

**Figure 4.** Overexpressed βPS/mys acts as a dominant-negative causing OR defects. (a–d) Tangential sections through equatorial regions of adult eyes with schematic presentations indicating ommatidial orientation below each section. Dorsal and ventral ommatidia are indicated with black and red arrows, respectively. Anterior is left. Ommatidia unscorable for orientation (due to photoreceptor loss) are marked with dots. βPS/Mys is expressed under *sevenless-Gal4* control (*sev-mys*) at the indicated temperature. (a) *sev > mys* at 25°C; such eyes are almost indistinguishable from *wt*, with ommatidia rotating largely at 90°. (b) *sev-mys* (2 copies) display OR defects. (c) *sev-mys* at 29°C (Gal4-driven expression is stronger at 29°C) causing mild rotation defects. (d) *sev-mys*; *if^{B2}*/+ (at 29°C). Heterozygosity for Integrin α-subunit, αPS2 (*if−*/+) enhances *sev > mys*, consistent with *sev > mys* acting as a dominant negative, by interfering with the balance between the α and β-integrin subunits. Note wide spread/distribution of rotation angles in backgrounds with 'defective' βPS/mys function. See figure 3b for quantification of genotype shown in b here.

**Table 1.** Genetic interactions with the *sev > mys* eye phenotype. A relevant subset of genetic interactions tested is shown for the genotype *UAS-mys; K25Gal4*. All interactions are in heterozygous conditions of indicated genotype. For all genotypes, *n* is at least 300 ommatidia from three to six independent eyes and standard deviation (s.d.) is between eyes. All interactions marked with asterisks have *p*-value. Information about the alleles used is available at http://flybase.bio.indiana.edu/.

| gene | % misrotated ommatidia ± s.d. | molecular feature of gene |
| --- | --- | --- |
| *UAS-mys; K25gal4* @25°C | 0.9 ± 1.1 | integrin β-subunit (βPS) |
| *UAS-mys/UAS-mys; K25gal4* @25°C | 52.1 ± 4.8* | integrin β-subunit (βPS) |
| for all genotypes below | | |
| *UAS-mys; K25gal4* @29 | 1.8 ± 0.3 | integrin β-subunit (βPS) |
| integrin subunits | | |
| *mys^1* | 27.4 ± 12.9* | integrin β-subunit (βPS) |
| *mys^{null}* | 36.4 ± 8.6* | integrin β-subunit (βPS) |
| *mew^{M6}* | 9.7 ± 5.3* | integrin α-subunit (αPS1) |
| *if^3* | 8.7 ± 0.1* | integrin α-subunit (αPS2) |
| *if^{B2}* | 10.7 ± 3.8* | integrin α-subunit (αPS2) |
| cytoskeletal adaptors/kinase | | |
| *vinc^1* | 16.3 ± 2.7* | vinculin |
| *stck^{3R-17}* | 9.0 ± 3.3* | pinch |
| *ILK^1* | 9.2 ± 3.8* | integrin-linked kinase |
| extracellular matrix components | | |
| *trol^{13}* | 8.0 ± 2.9* | Perlecan |
| *wb^{09437}* | 5.9 ± 2.6* | Laminin-α1,2 |
| *tig^{A1}* | 4.1 ± 0.9* | Tiggrin |
| *sdc^{10608}* | 7.2 ± 3.2* | Syndecan |

(Continued.)

royalsocietypublishing.org/journal/rsob    Open Biol. 9: 190148

**Table 1.** (Continued.)

| gene | % misrotated ommatidia ± s.d. | molecular feature of gene |
|------|-------------------------------|---------------------------|
| core PCP factors/N signalling | | |
| $fz^{P21}$ | 0.3 ± 0.5 | Frizzled receptor |
| $fz^{R52}$ | 1.9 ± 2.4 | Frizzled receptor |
| $stbm^{null}$ | 12.5 ± 5.6* | Strabismus/Van Gogh (Vangl in mammals) |
| $dsh^1$ | 2.5 ± 1.1 | Dishevelled/Dsh (Dvl in mammals) |
| $dgo^{380}$ | 1.8 ± 1.5 | Diego/Inversin/Diversin |
| $N^{55e11}$ | 2.4 ± 1.1 | Notch receptor |
| $Dl^{rev10}$ | 1.6 ± 0.9 | ligand for Notch (Dll in mammals) |
| other OR-related | | |
| $RhoA^{720}$ | 6.7 ± 1.6* | GTPase |
| $dRok^2$ | 20.7 ± 4.9* | Rho kinase |
| $zip^1$ | 9.5 ± 4.4* | myosin-II heavy chain |
| $sqh^{PL91}$ | 21.7 ± 2.8* | myosin-II regulatory light chain |
| $chic^{01320}$ | 15.2 ± 1.9* | profilin |
| $svp^{7842}$ | 2.2 ± 0.3 | nuclear hormone receptor |
| $Egfr^{top1}$ | 1.9 ± 0.8 | EGF-receptor/RTK |
| $aos^7$ | 1.7 ± 1.1 | EGF-receptor ligand |
| $nmo^P$ | 2.4 ± 1.5 | Nlk, MAPK family kinase member |
| $nmo^{DB}$ | 2.1 ± 1.6 | Nlk, MAPK family kinase member |
| $arm^{XM19}$ | 2.0 ± 1.0 | β-catenin |
| $shg^{P34-1}$ | 2.3 ± 1.4 | E-cadherin |
| $sca^{BP1}$ | 2.1 ± 0.7 | fibrinogen-related |
| for all genotypes below | | |
| $UAS\text{-}Torso^D\text{-}bPScyto;$ K25gal4 /+ | 10.0 ± 2.5 | dominant activated integrin β-subunit (βPS) |
| $mys^{null}$ | 19.9 ± 3.1* | integrin β-subunit (βPS) |
| $wb^{09437}$ | 8.1 ± 1.0 | laminin-α1,2 |
| $ILK^1$ | 31.9 ± 6.8* | integrin-linked kinase |
| UAS-Rho-IR | 19.1 ± 4.5* | Rho GTPase |
| $dRok^2$ | 27.6 ± 8.3* | Rho kinase |
| $fmi^{E45}$ | 35.2 ± 6.0* | Protocadherin with 7TM region |

*$p < 0.005$.

## 3.3. Integrin βPS/mys is localized to cellular membranes at the periphery of ommatidial clusters

As loss of integrin function causes OR defects, we wished to determine the expression pattern and localization of integrins in developing eye discs during patterning and OR. Third instar larval eye discs were stained with a βPS/mys-specific antibody and analysed by confocal microscopy (figure 5). First, βPS/mys was detected basally throughout the eye disc (figure 5d). Second, posterior to the furrow starting at around column 5, βPS/mys became enriched laterally in cell membranes on the outside of individual ommatidial pre-clusters basal to the adherens junctions, which were marked by E-cadherin staining (figure 5a,b,d). By column 9–10, βPS/mys was also enriched laterally in photoreceptors R1/R6 within each ommatidial cluster (electronic supplementary material, figure S4).

It is important to note that at these stages, epithelial integrity was preserved in the loss of βPS/mys background (figure 5a,c,d). We did not detect any defects in apical–basal polarity in eye imaginal discs, nor did it seem to affect general ommatidial cluster integrity in third instar discs, as all apical markers tested (e.g. E-cadherin; figure 5a,d), basolateral cell membrane markers (FasIII; figure 5c,c') and nuclear markers (Elav, figure 5d) appeared unaffected.

Taken together, these data demonstrate that integrin localization is dynamic during eye development and OR. It is noteworthy that integrins are enriched laterally surrounding each photoreceptor cluster (or in other words: on the outside cell surfaces of R cells, all around any given cluster), forming a goblet-like 'container' in which the cluster is positioned (figure 5d).

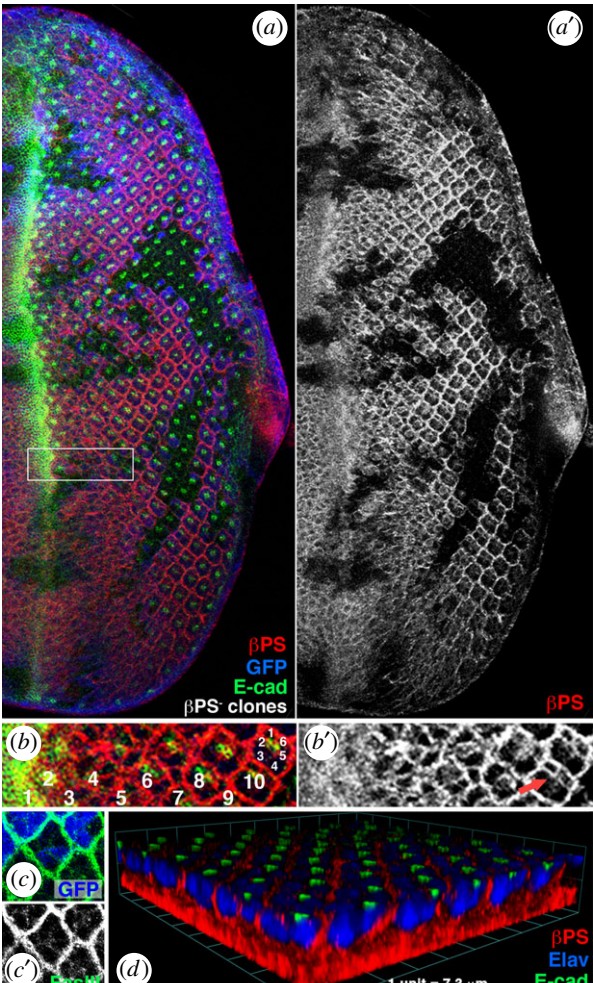

**Figure 5.** Localization of Mys/βPS–integrin during eye disc patterning. All panels show confocal images of third instar eye discs. Anterior is left and dorsal is up. Integrin βPS subunit, visualized by a specific antibody (red in *a–c*, and single channel in *a′*, *b′* and *c″*), is highly enriched at membranes that are at the periphery of the developing ommatidial clusters, basal to adherens junctions (marked by DE-cadherin, green in *a*, *b* and *d*), and on the basal sides of each cell. (*a–c*) *mys¹* clones (mutant for βPS–integrin subunit) are marked by the absence of GFP (blue in *a* and *c*, single channel in *c′*). (*b*) Enlargement of boxed area in (*a*). Approximately 9–11 columns posterior to the morphogenetic furrow, βPS/Mys is also detected around the cells of the R1/R6 pair (red arrow in *b′*). Numbers 1–10 mark the respective columns posterior to MF, the respective R cells are labelled with small white numbers (1–6) in most posterior cluster shown. See electronic supplementary material, figure S4 for more on R1/R6 expression. (*c,c′*) The loss of βPS/Mys does not cause apical–basal polarity defects, nor does it affect precluster integrity in third instar eye discs, as apical markers (DE-cadherin, green in *a*) and markers for the basolateral plasma membrane (FasIII, green in *c*; single channel in *c′*) are unaffected. (*d*) Three-dimensional rendering of confocal stacks showing the βPS/Mys localization (red) baso-laterally, having a 'basket'-like appearance around each cluster. Photoreceptor nuclei are marked with anti-Elav (blue) and DE-cadherin (green) marks the apical adherens junctions.

## 3.4. Integrins mediate a link between cytoskeletal components and the ECM during rotation

The phenotype of a single copy *sev > mys* (dominant negative, see above) at 29°C allows for genetic screening of potential interacting genes, as it is enhanced by loss of one gene copy of integrin subunits (βPS, αPS1 or αPS2; figure 4 and

table 1). To gain insight into how integrins may be linked with other cellular functions during OR, we tested candidate genes for potential comparable interaction(s) with the *sev > mys* phenotype. Using this approach, we assayed both cytoplasmic/cytoskeletal factors and components of the ECM (table 1; schematic presentation of some components of the ECM–integrin–actin cytoskeleton linkage are shown in figure 6*a*).

Reducing endogenous levels of the cytoskeletal adaptor proteins Vinculin (*vinc¹*) or Pinch (*steamer duck, stck³ᴿ⁻¹⁷*) enhanced the *sev > mys* phenotype (table 1). Similar results were obtained with a mutation in integrin-linked kinase (*ILK¹*; table 1). Furthermore, profilin (*chickadee, chic⁰¹³²⁰*), an actin monomer-binding protein, strongly interacted with *sev > mys* (table 1). The absence of profilin causes actin-dependent processes including cell motility and polarized growth to fail [63]. We also tested the motor protein non-muscle myosin-II for dominant interactions. Both the heavy chain (*zipper; zip⁰²⁹⁵⁷*) and regulatory light chain (*spaghetti squash; sqhᴾᴸ⁹¹*) of myosin-II enhanced the *sev > mys* induced rotation defects (table 1). Taken together, these data indicate that a physical link of βPS/integrin to the actin cytoskeleton and motor proteins is functional and an important regulatory mechanism during OR.

To address whether components of the ECM are required in OR, we tested reduced levels of Perlecan (*terrible reduced optic lobes, trol¹³*), Laminin-α1,2 (*wing blister, wb⁴¹ʸ⁸*) and Tiggrin (*tigᴬ¹*). All three enhanced the *sev > mys* rotation phenotype (table 1). Perlecan, a heparan sulfate proteoglycan, binds to other ECM molecules including laminins. Laminin-α1/2 and Tiggrin, in turn, directly bind to βPS integrins [64,65]. Interestingly, we also observed a dominant enhancement of *sev > mys* by Syndecan mutants (*sdc¹⁰⁶⁰⁸*; table 1). A considerable amount of cell culture data suggest cross-talk between integrins and syndecans, which are cell surface molecules with several heparan sulfate side chains that can act as a co-receptor for integrins (reviewed in [66]). Taken together, these data suggest a complex network of interactions between the ECM and integrins in regulating OR.

The observed interactions between *sev > mys* and *trol* or *wb* prompted us to directly investigate the role of these ECM proteins during OR. Homozygous animals for hypomorphic alleles of *trol* have blistered wings, a phenotype resembling integrin and laminin mutant phenotypes, and rough eyes [67]. Our analysis of eye sections of *trol¹³* escaper flies revealed rotation defects (figure 6*b*), providing additional evidence for an ECM requirement in OR.

Laminins consist of a long α-chain and shorter β and γ-chains arranged in an umbrella-like structure and only α, β and γ-heterotrimers give rise to functional proteins [68,69]. *Drosophila* has two Laminin-α subunits, α1/2 and α3/5, encoded by *wb* and *lamininA* (*lanA*), respectively. Heteroallelic mutant combinations of *lanA* cause rough eyes with ommatidial orientation defects [70]. *wb* is required for cell adhesion and migration during embryonic and imaginal disc development and many phenotypes are shared with those of integrin mutants [64]. As for integrins, adult eye sections of *wb* mutant clones have disorganized rhabdomeres precluding analysis of rotation defects ([64] and our data). Thus, we analysed loss-of-function clones in third instar eye discs. A null allele, *wb⁴ʸ¹⁸*, affected OR in a non-autonomous fashion, with many ommatidia affected, both within mutant tissue and in adjacent wild-type regions (figure 6*c–c″*). The

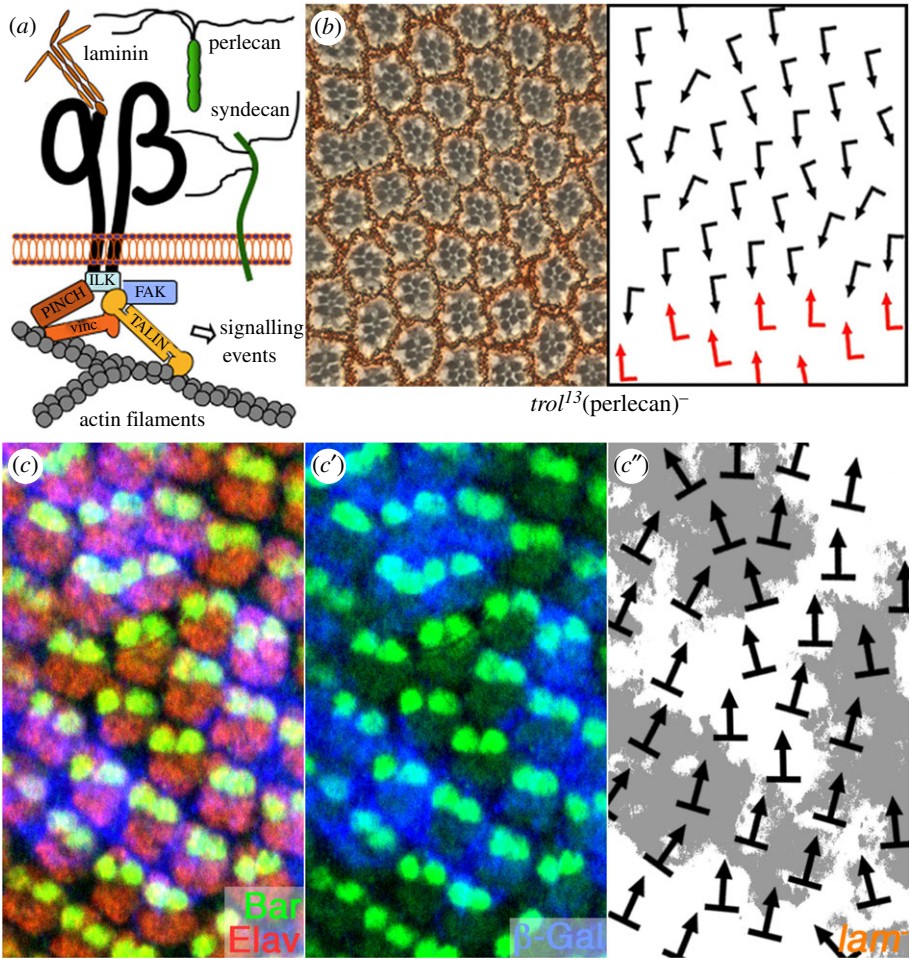

**Figure 6.** ECM components are required for OR. (*a*) Schematic presentation of selected components and factors associated with integrin-mediated adhesion and signalling. Integrin heterodimers (α and β chain shown in black) and components of the ECM are shown in the top half of the panel, components mediating integrin signalling in the cytoplasm are displayed in the lower half of the panel (ILK, integrin-linked kinase; FAK, focal adhesion kinase; vinc, vinculin). (*b*) Perlecan is required for correct OR. Tangential sections through adult eyes of the hypomorphic *trol*¹³ allele reveal OR defects. Ommatidial orientation is indicated with arrows (schematic to the right); arrows are as in figure 4. (*c–c'*) Confocal image of the third instar eye imaginal disc showing clones mutant for a laminin-α chain (*wb*⁴ʸ¹⁸). Mutant tissue is marked by lack of β-gal staining (blue); photoreceptors are marked by anti-Elav (red) and anti-Bar (green), highlighting the R1/R6 pair reflecting ommatidial orientation. Ommatidia within mutant and *wt* tissue are misrotated indicating the non-autonomous function of secreted Laminin. (*c''*) Schematic of ommatidial orientation from *c–c'*: white areas represent *wb/laminin-α* mutant tissue; black arrows indicate rotation angles (as deduced from R1/R6 cell arrangement relative to equator).

enhancement of the *sev > mys* phenotype by *perlecan*, *tiggrin* and *laminin* mutations (table 1), and the rotation defects seen in *lanA* mutants [70], *wb* and perlecan/*trol* mutant tissue (figure 6, this work), suggest that ECM components are required for the integrin-based aspect of OR.

To gain further insight into the role of ECM proteins in OR, we also compared the expression and localization of LanA with βPS/mys in eye imaginal discs during ommatidial patterning and OR. Strikingly, LanA is localized in a similar pattern as the one detected with anti-βPS/mys antibodies (electronic supplementary material, figure S5). Laminin is detected in the space surrounding clusters and in between ommatidial clusters, besides its enrichment in the basal lamina (electronic supplementary material, figure S5). As the cells and clusters are tightly packed in the eye disc, the localization patterns of βPS/mys and Laminin appear very similar and partially overlapping with the resolution of regular confocal microscopy (electronic supplementary material, figure S5). The pattern of βPS/mys does not depend on the presence of LanA, as in *LanA* mutant clones, βPS/mys appears unaffected; this is consistent with βPS/

mys interacting with several components of the ECM. The similar localization patterns of an ECM component (LanA) and βPS/mys support the notion that integrins interact with ECM components during the rotation process.

## 3.5. PCP genes genetically interact with integrins during the rotation process

We next analysed potential input from PCP genes into the function of integrins in OR. Both *strabismus/Van Gogh* (*Vang*ˢᵗᵇᵐ⁶) and *RhoA* (*RhoA*⁷²⁰) enhanced the rotation defects, suggesting that integrin function is (in part) linked to PCP signalling (table 1; note that *fz* did not show a genetic interaction with *sev > mys*). In addition, we observed a strong interaction between *sev > mys* and *Drosophila Rho kinase* (*dRok*²), a downstream effector of RhoA signalling (table 1, and below).

During the genetic interaction studies using *sev > mys*, we noticed that in several cases, no progeny of the expected genotype was recovered due to pupal lethality, likely due to

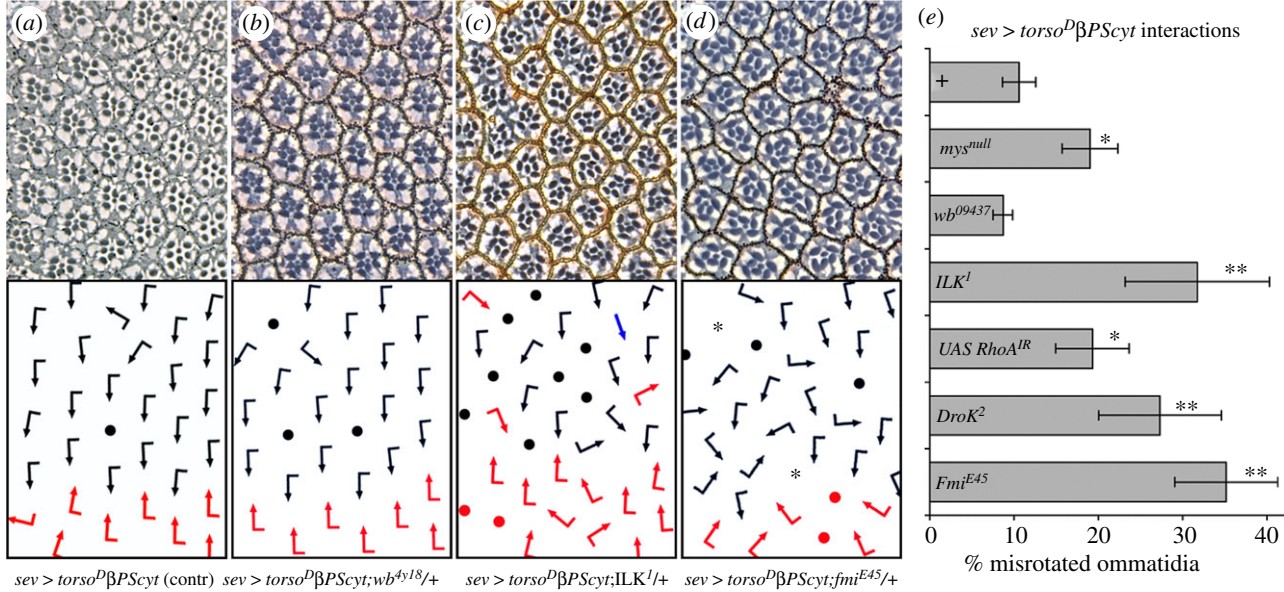

**Figure 7.** A dominant-negative integrin-signalling chimera results in rotation defects. (*a–d*) Tangential sections through equatorial regions of adult eyes, with schematics indicating ommatidial orientation below each section. Arrows are as in figure 4. Unscorable ommatidia (due to loss of R cells) are marked with dots. The chimeric $torso^D\beta PS_{cyt}$ protein is expressed under *seveless-Gal4* control ($sev > torso^D\beta PScyt$) at 29°C. (*a*) $sev > torso^D\beta PS_{cyt}$: note approximately 10% misrotated ommatidia, (*b*) $sev > torso^D\beta PS_{cyt}$; $wb^{EY18}$/+. (*c*) $sev > torso^D\beta PS_{cyt}$; $ILK^1$/+. (*d*) $sev > torso^D\beta PS_{cyt}$; $fmi^{E54}$/+. Note that in addition to rotation defects, reducing the levels of these genes also caused an increase in R-cell number defects (loss of R cells—dots—and R7 to R1–6 transformation—asterisk in *d*—in schematic). (*e*) Quantification of $sev > torso^D\beta PS_{cyt}$interactions: * and ** indicates *p*-values < 0.05 and <0.005, respectively (with Student's *t*-test). Whereas no effect is seen in a *wb/laminin* heterozygous background as expected (*b*), reduced dosage of *ILK* (*c*) and *fmi* (*d*) enhances the $sev > torso^D\beta PS_{cyt}$ phenotype.

leakiness in the *sev-Gal4* expression system [71]. To bypass this, we used an integrin-chimera that affects cellular signalling without interfering with adhesion: $Torso^D\beta PS_{cyt}$, a chimera with the extracellular domain of Torso carrying a mutation causing constitutive dimerization [72] attached to the cytoplasmic tail of βPS/mys [73]. This leads to effector clustering independently of a 'ligand', the ECM, and allowed us to recover several combinations that were lethal with *sev > mys*.

Expression of $UAS-Torso^D\beta PS_{cyt}$ (under *sev-Gal4*) revealed that Torsoβ$PS_{cyt}$ functions as a dominant-negative integrin receptor (comparable to *sev > mys*), as rotation defects were enhanced in *mys* heterozygous backgrounds (figure 7*a,e* and table 1). $sev > torso^D\beta PS_{cyt}$ was also enhanced by heterozygosity of ILK (integrin-linked kinase), consistent with the *sev > mys* data (figure 7*c,e* and table 1). By contrast, no interaction was observed with the ECM component laminin ($wb^{41Y8}$), as expected in the context of a 'ligand-independent' receptor (figure 7*b,e* and table 1).

With $sev > torso^D\beta PS_{cyt}$, we uncovered a strong dominant enhancement with *flamingo* ($fmi^{E45}$/+), causing a fourfold increase in misrotated ommatidia (figure 7*d,e*). Fmi, a seven-pass transmembrane cadherin and a core Fz-group PCP factor [5,74–76], also genetically interacts with Egfr/Ras signalling during OR [21,22]. These data support a link between integrins and core PCP genes in OR. Co-overexpression of $sev > torso^D\beta PS_{cyt}$ with a UAS construct lowering endogenous RhoA levels (through RNA interference via $UAS-RhoA^{IR}$) led to a specific enhancement of the rotation defects (figure 7*e*; confirming the enhancement of $sev > mys$ by $RhoA^{720}$ and $dRok^1$ heterozygosity; table 1).

Taken together, these data are consistent with an important role of intracellular signalling of integrins during OR, and they suggest a cross-regulatory interaction between

integrins and the core Fz/PCP factors and their effectors, including RhoA and dRok.

## 3.6. *dRok* causes an upregulation/accumulation of βPS–integrin and impairs rotation

Two of the strongest enhancers of *sev > mys* were alleles of *dRok* and non-muscle *myosin-II regulatory light chain* ($sqh^{PL91}$) (table 1). dRok has previously been reported to function downstream of Dsh/RhoA in PCP establishment, and in the eye predominantly affects OR [23].

To explore the function of PCP signalling via dRok in the integrin-based OR function, we expressed a constitutively activate dRok in clones in third instar eye discs (using the Flip-out technique). Strikingly, this revealed that the levels of βPS/mys were strongly increased in cells in which activated dRok was expressed, both posterior and anterior to the MF (figure 8). Particularly, during OR posterior to the furrow, an increase in integrins was evident around each affected cluster (orange arrows in figure 8).

Taken together with the genetic interactions, these data suggest that the core PCP factors might influence integrin levels and/or accumulation via Rho-dRok signalling.

## 4. Discussion

OR in the *Drosophila* eye imaginal disc is a highly organized and coordinated movement of groups of cells. How this movement is regulated is largely unknown. We demonstrate that an integrin/ECM-mediated link is needed for precise and coordinated rotation. Furthermore, our data suggest that integrin membrane accumulation is regulated via

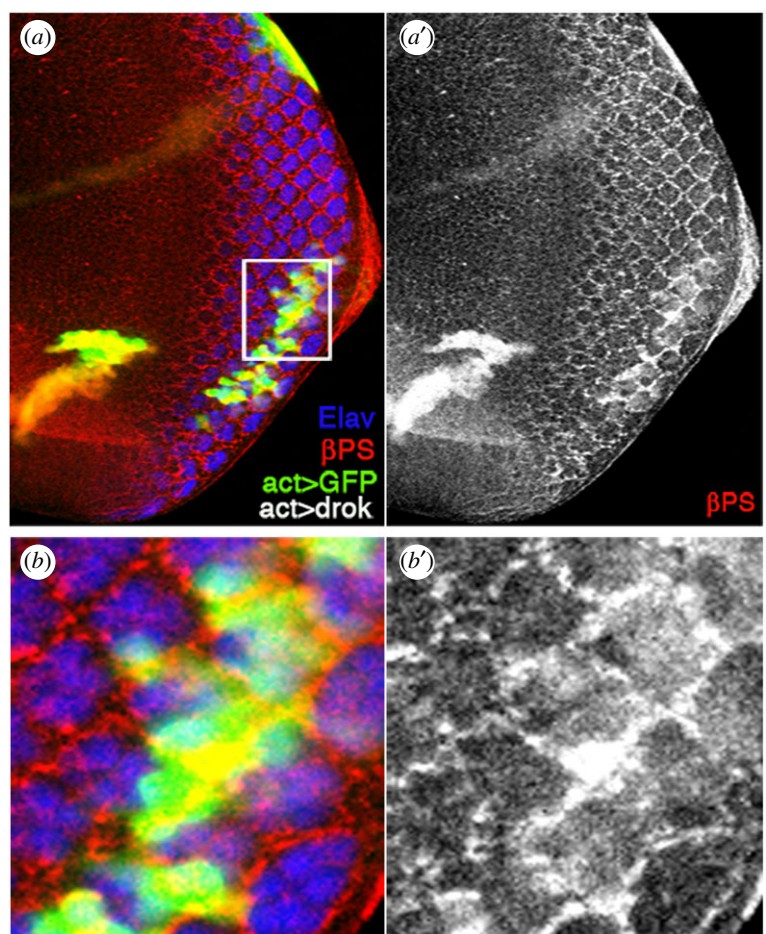

**Figure 8.** Activated Rho kinase causes accumulation of βPS/integrin levels. Confocal images of the third instar eye imaginal disc with activated dRok expression clones (generated via the Flip-out technique) marked by simultaneous GFP expression (green). R-cell neurons are labelled by anti-Elav (blue), βPS/Mys integrin staining is in red (single channel in monochrome panels *a′* and *b′*). (*a,a′*) Both posterior and anterior to the furrow, βPS/Mys levels are upregulated in dRok-expressing clones (clone anterior to furrow is 'overlayed' by a peripodial membrane clone: thus, clone boundary and integrin staining do not appear exactly coincident, since dRok in peripodial membrane cells does not cause βPS/Mys upregulation). (*b,b′*) Enlargement of boxed area posterior to the furrow in (*a*); note that dRok-dependent βPS/Mys accumulation is mainly detected at the outside membranes of preclusters (where 'rotation takes place') as in wild-type (see figure 5 for wild-type details for comparison).

RhoA–dRok signalling, which are effectors of the core Fz/PCP pathway.

## 4.1. The role of integrins, the ECM and dRok in ommatidial rotation

Our data indicate that loss of integrin function during OR impairs the normal rotation process. In integrin mutant tissue, ommatidia rotate asynchronously with many rotating faster and others slower, leading to an imprecise rotation in more mature clusters in both the eye disc and in adult eyes, when compared with the precise 90° rotation in wild-type clusters. These data suggest that integrins are involved in controlling the pace of rotation.

OR is an 'effector process' in PCP establishment. The direction of rotation is determined by R3/R4 fate specification and a link between core PCP signalling and rotation via effector regulation of cell adhesion and cytoskeletal elements has been suggested [17,18,24], but it is far from being well understood. RhoA is a downstream component of core PCP signalling, affecting both R3/R4 fate specification and subsequent rotation [77], while the RhoA effector dRok specifically affects OR, but not R3/R4 fate decisions [23]. We detected interactions (enhancement) of *sev > mys* with

*RhoA*, *dRok* and, also, *zip/Myo-II* and myosin-II regulatory light chain (*sqh/MRLC*), suggesting that these components function together with integrins during OR.

RhoA/dRok signalling regulates MRLC/Sqh phosphorylation [23,78,79], promoting the formation of actin stress fibres and the generation of contractile force, which drives cell movement and cell shape changes [80,81]. Expression of activated dRok in ommatidial preclusters during OR leads to an aberrant, increased accumulation of βPS/integrin at cell membranes (figure 8). This effect could reflect 'endpoints' of newly formed stress fibre-related structures and/or the formation of integrin-rich focal complexes at the plasma membrane. In vertebrates, αβ1 integrins, which are equivalent to *Drosophila* αβPS integrins, mediate the turnover of focal contacts and actin stress fibres, and consequently, the polarized movements of αβ1 integrin mutant cells are also defective [82]. Rho-associated kinases (Rok) can both positively and negatively regulate cell motility, depending upon context [83–85]. The link between integrins and dRok reveals a potential mechanism by which PCP signalling could impact and affect the OR process.

An intriguing aspect of rotation is that cells within clusters stay tightly connected as a unit, while rotating past neighbouring inter-ommatidial epithelial cells and the ECM. Ommatidia rotate 90° towards the equator, clockwise

in the dorsal half and counterclockwise in the ventral half of the eye. Although integrin-dependent cell migration and associated assembly and de-assembly of focal complexes are well studied, the majority of these studies have been performed in two-dimensional tissue culture conditions on matrix-coated surfaces, a somewhat artificial milieu. Analysing tumour-cell motility in three-dimensional matrices has suggested that Rho signalling through Rho kinase promotes a rounded, bleb-associated mode of motility [83] that might resemble OR rotation.

## 4.2. PCP signalling and integrins

OR is the final step of PCP-mediated events during eye patterning and also the least understood process. Rotation is a precisely coordinated movement, and although R3/R4 fate specification controls direction of rotation, the mechanistic aspects of the actual rotation process are basically unknown. What directly mediates the initiation remains elusive. Whereas the *nmo* kinase promotes rotation [16–18], the role of Egfr signalling is less well defined [20–22]. We did not observe genetic interactions between βPS/*mys* and *nemo* and cadherins, or components of the Egfr pathway. This was consistent with previous studies demonstrating that components of Egfr signalling do not interact with integrins during OR [20,21], suggesting that Egfr-signalling, cadherin-based adhesion and integrins have different roles in controlling the process.

The PCP gene *fmi* (an atypical cadherin) interacts with both the Egfr pathway and with integrins during OR. Fmi localization is disturbed in *argos* ($aos^{rlt}$) mutant eye discs [21,22], but although *Fmi* is a core PCP gene, R3/R4 fate specification was not affected in $aos^{rlt}$. However, heterozygosity for *fmi* strongly enhanced the $sev > torso^D \beta PS_{cyt}$ rotation phenotype, and removing one *fmi* copy in a $sev > mys$ background was even lethal, although Fmi levels appeared unaffected in βPS/*mys* mutant tissue. Heterozygosity for *stbm/Vang* also enhanced the βPS/*mys* rotation phenotypes (table 1). Stbm/Vang specifies the R3/R4 fate decision and rotation is delayed in *stbm/Vang* mutant eye discs [86], and Stbm/Vang has been shown to serve as the molecular link between core PCP complexes and the Nmo kinase, which links these to E-cad-mediated adhesion [18,19]. As the integrin-associated OR phenotypes do not interact with Egfr-signalling components or Nmo (table 1), these data suggest a complex interplay of at least two systems downstream of core PCP signalling in the regulation of adhesion, involving independent regulation of cadherin and integrin-based adhesion (also below).

In this context, it is worth noting that the OR 'specific' genes/effects characterized thus for can be grouped into two categories. The first set appears to promote rotation and the respective factors include E-cad and Nmo, for example, and accordingly in the respective *nmo* and *shg* mutants OR largely does not proceed [16–18]. By contrast, mutations associated with the Egfr pathway, including its ligand Argos or downstream Egfr–Ras signalling effectors, cause a largely randomized rotation with respect to OR angles [18,20,21]. Although these defects resemble superficially the defects we observe in integrin mutant backgrounds, the Egfr-associated defects have been suggested to be mediated by cadherin-based adhesion regulation [18,20,21]. Taken together, our data reported here add a third category that is required for precise, coordinated OR, but that is not mediated by Egfr signalling. The limitations of this and other current studies are the inherent reliance on simplified phenotypic observations (in a very complex morphogenetic process) and potential genetic interactions based on these simplified phenotypic features. As such, the complicated links likely to occur between different aspects of cell adhesion and cell signalling are not yet adequately covered.

Besides OR, other cell motility processes function downstream of (or as functional read-outs of) PCP signalling. During vertebrate convergent extension in gastrulation, mesenchymal cells move towards the midline (convergence), followed by their intercalation, causing the extension of the body axis (reviewed in [87]). A role for integrins and ECM components in convergence extension during *Xenopus* gastrulation is supported [52,88,89]. As interactions among core PCP factors and their link to cell adhesion regulation are well conserved (reviewed in [4,5,74–76,90,91]), it is thus likely that a PCP link to integrins is also a conserved feature associated with many if not all PCP-regulated cell motility processes.

**Data accessibility.** All data supporting this article have been uploaded as part of the manuscript or electronic supplementary material.

**Authors' contributions.** M.T., K.G. and M.M. conceptualized and designed the study, analysed all data and wrote the manuscript; M.T. and K.G. generated the *Drosophila* strains and majority of the data and acquired, analysed and interpreted data; G.M.C., M.-l.C. and J.S. acquired, analysed and interpreted subsets of the data; G.M.C. and M.M. assessed and commented on the data, provided intellectual and biological context-related input and revised the manuscript to obtain its final version for publication. M.M. funded the project and provided coordination and supervision throughout.

**Competing interests.** We declare we have no competing interests.

**Funding.** This work was supported by an NIH/NEI grant no. RO1 EY13256 to M.M. M.T. was supported by the Wenner-Gren Foundation, the Royal Swedish Academy of Sciences and The Sweden-American Foundation. Confocal laser scanning microscopy was performed at the ISMMS-Microscopy Core Facility supported by the Tisch Cancer Institute grant no. P30 CA196521 from the NCI.

**Acknowledgements.** We are grateful to K. Basler, N. Brown, D. Brower, T. Bunch, L. Luo, K. Saigo, F. Schöck, the Bloomington Stock Center and the DSHB for *Drosophila* strains and antibodies. K.G. and M.M. would like to thank C. Samakovlis for providing infrastructure for *Drosophila* experiments, and K.G. is most grateful to C. Betsholtz for his continued support. We thank N. Founounou and all other members of the Mlodzik lab, and S. Johansson for discussion and helpful comments on earlier drafts of the manuscript.

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
