## [Reviewer comments · Open Biology]

Review History

RSOB-19-0148.R0 (Original submission)

Review form: Reviewer 1

Recommendation

Accept with minor revision (please list in comments)

Scientific importance: Is the manuscript an original and important contribution to its field?

Good

General interest: Is the paper of sufficient general interest?

Acceptable

Quality of the paper: Is the overall quality of the paper suitable?

Good

It is a condition of publication that authors make their supporting data, code and materials available - either as supplementary material or hosted in an external repository. Please rate, if applicable, the supporting data on the following criteria.

Is it accessible?

N/A

Is it clear?

N/A

Is it adequate?

N/A

Do you have any ethical concerns with this paper?

No

Comments to the Author

Nicely written introduction very wide ranging and perhaps a bit too long. It emphasises how integrins are functionally important in many processes as they not only are agents to attach a cell to the ECM they are also involved in intercellular signalling, partially indirectly, by “promoting adhesion to the ECM so that cells are kept close to the source of signals”. No surprise then that flies lacking integrins are devastated. So, the approach here, a familiar and well justified one, is to make small clones lacking integrin function *mys*-, so that late requirements can be studied in some isolation. The focus of attention is on ommatidial rotation, a late and interesting process where small groups of cells rotate en bloc. This local loss of *mys* affects rotation in the imaginal discs, while by the adult eye stage the *mys*- clones have become a complete mess.

More gentle interference with the *myospheroid* gene (localised and late overexpression under sevenless control *sev>mys*) has a milder consequence causing defects of rotation and other problems with the adult eye pattern. These flies can be used as a sensitised background to test for the role of other proteins.

Evidence for link between integrins and ECM being required for rotation is reinforced when other proteins involved in attachment to the ECM are also removed – the relatively weak mutant phenotype induced by overexpression of *mys* is aggravated. The authors then build the evidence that attachment to the ECM is instrumental in rotation by removing these other proteins in clones and finding rotation defects. These methods demonstrate interactive requirement, they are not so good at working out what the integrins and the other genes do, particularly as the effect of overexpression is itself a mutant phenotype, and disturbing rotation is relatively unspecific. Nevertheless the work raises some interesting if only suggestive ideas.

The authors then turn their attention to PCP using the overexpression of *sev>mys* flies as a background. The key protein in PCP is Frizzled and its removal does not work on this assay. Surprising..... maybe the authors should emphasise in the discussion that this jigsaw puzzle piece does not fit. However, the loss of Vang, another PCP protein of less centrality than Fz, does increase rotation defects. Even halving the amount of Flamingo has an effect in some special flies overexpressing *mys* and this adds to the argument of an involvement of PCP in the rotation process.

There are some relevant experiments that are missing, for example do *nemo* and E-cad mutations increase the phenotype of *sev>mys*? What about testing the interaction the other way around? – are PCP defects increased in a *myo/+* background? Do *myo/+* flies affect PCP non-autonomous signalling?

Overall the authors have done a careful investigation of genetic interactions to build an argument that some PCP proteins are involved, with the ECM, in rotation. The authors should make it

clearer that the investigation uses a rather blunt scalpel: these are compromised eyes, the experiments combine one genetic sickness with another to deduce mutual dependency. Yes they use the conventional experimental language of developmental genetics but the arguments that they make are not watertight and some recognition of this should be made. Also, we are not much wiser about what the wildtype mechanism is. Nevertheless I think the results are useful, they widen the picture of rotation in an interesting way. Moreover, the whole paper is clear and well written, the technical quality of the work is high and well presented with beautiful images, detailed and convincing analysis of ommatidial orientations and we recommend that the paper should be accepted subject to some revision.

About Figure 5:

The authors claim that during a short period there is an accumulation of myo protein around R1 and R6. However, this is difficult to see in the figure. The authors should consider using a better image. Even if this finding is correct, the relevance is not clear, is the reader being taken on a rather foggy trip for no purpose?

Review form: Reviewer 2

Recommendation

Accept as is

Scientific importance: Is the manuscript an original and important contribution to its field?

Excellent

General interest: Is the paper of sufficient general interest?

Excellent

Quality of the paper: Is the overall quality of the paper suitable?

Excellent

It is a condition of publication that authors make their supporting data, code and materials available - either as supplementary material or hosted in an external repository. Please rate, if applicable, the supporting data on the following criteria.

Is it accessible?

Yes

Is it clear?

Yes

Is it adequate?

Yes

Do you have any ethical concerns with this paper?

No

Comments to the Author

Thuveson and colleagues report on the function of integrins and the ECM during *Drosophila* eye development, in particular during ommatidial rotation. They use loss of function clones and genetic interactions with overexpressed dominant negative forms of integrins to show that

integrins are essential for proper ommatidial rotation and interact with planar cell polarity signaling components, which are known to be important for ommatidial rotation. They show that integrin localization changes dynamically during rotation and accumulates on cell membranes facing outside of ommatidial clusters.

This is a beautifully illustrated and well-presented paper describing specific in vivo functions for integrins and ECM components. The specific localization of integrins and how this is affected by Drok activation is particularly interesting and provides new insights into the difficult question of how ECM-integrins cause specific effects. I have only a minor suggestion: the rotation defects in integrin mutant clones is clear, however, the quantification as presented does not really illustrate the randomness and disorganization of the tissue. Maybe this could be illustrated by tabulating the difference in angle between neighboring clusters.

Review form: Reviewer 3

Recommendation

Accept with minor revision (please list in comments)

Scientific importance: Is the manuscript an original and important contribution to its field?

Good

General interest: Is the paper of sufficient general interest?

Good

Quality of the paper: Is the overall quality of the paper suitable?

Good

It is a condition of publication that authors make their supporting data, code and materials available - either as supplementary material or hosted in an external repository. Please rate, if applicable, the supporting data on the following criteria.

Is it accessible?

Yes

Is it clear?

Yes

Is it adequate?

No

Do you have any ethical concerns with this paper?

No

Comments to the Author

This paper explores the role of integrins in eye development, in particular its role in ommatidial rotation. The authors show that integrins are needed for the normal progression of ommatidial rotation, using a clones of mutant allele of betaPS/mys, and overexpression of betaPS/mys constructs that cause dominant negative effects. The authors explore the pathway of this integrin function using genetic interactions, which show that reduction of a number of components of the integrin machinery, actin machinery and PCP machinery are able to enhance the dominant

negative integrin construct phenotypes. This is a useful data set, but does not contain any surprises, and also falls short of providing a clear mechanism for how integrins contribute to ommatidial rotation. The value of the data set could be improved by 3 straightforward revisions.

1. The paper seems to be hinting that loss of integrin is unusual in that rotation is accelerated, whereas other mutations that impair rotation just reduce the rate. The authors need to be more explicit about this potential key difference and describe which mutants show accelerated rotation or retarded rotation, both in the published literature and the experiments presented in this paper. For example is there also acceleration in the perlecan and wing blister mutants shown in Fig. 6? Is this a unique feature of loss of integrin mediated adhesion? This might imply that rotation is primarily driven by lateral movement, and the basal attachment provides useful resistance to regulate the rate of movement. Is it possible to image the cell shape from apical to basal, and see whether the rotation is ahead at the top or bottom of the cell? This relates to recent work showing that convergent extension can be lead by junction exchange either apically or basally.

2. Expand Table 1 to include mutations that do not enhance the phenotype. Just showing mutations that enhance the phenotype leaves open the possibility than any mutation will do so. Showing the specificity of this enhancement is therefore valuable, and it is just as interesting to know the mutations that do not enhance as those that do enhance.

3. Allow a better comparison between the data in Table 1 and Fig 7.E. Using a different dominant negative construct for the genetic interaction experiments is a very useful addition, and the differences are especially interesting. However both data sets need to be displayed in the same way so we can directly compare them. Either both in a Figure or both in Table 1.

Minor errors

Intro.

p4 Cooper and Bray reference should be the 1999 one, not the 2000 one

p6, 2nd para. Beta-nu is not only functional in the midgut; there is evidence the beta-nu functions in haematopoietic cells, e.g Nagaosa et al., 2011 DOI10.1074/jbc.M110.204503

p13,14 the symbol for lamininA is LanA; lama is a different gene

Decision letter (RSOB-19-0148.R0)

11-Jul-2019

Dear Professor Mlodzik

We are pleased to inform you that your manuscript RSOB-19-0148 entitled "Integrins are required for synchronous ommatidial rotation in the Drosophila eye linking Planar Cell Polarity signaling to the extracellular matrix" has been accepted by the Editor for publication in Open Biology. The reviewer(s) have recommended publication, but also suggest some minor revisions to your manuscript. Therefore, we invite you to respond to the reviewer(s)' comments and revise your manuscript.

Please submit the revised version of your manuscript within 14 days. If you do not think you will be able to meet this date please let us know immediately and we can extend this deadline for you.

- 1) A text file of the manuscript (doc, txt, rtf or tex), including the references, tables (including captions) and figure captions. Please remove any tracked changes from the text before submission. PDF files are not an accepted format for the "Main Document".
- 2) A separate electronic file of each figure (tiff, EPS or print-quality PDF preferred). The format should be produced directly from original creation package, or original software format. Please note that PowerPoint files are not accepted.
- 3) Electronic supplementary material: this should be contained in a separate file from the main text and meet our ESM criteria (see <http://royalsocietypublishing.org/instructions-authors#question5>). All supplementary materials accompanying an accepted article will be treated as in their final form. They will be published alongside the paper on the journal website and posted on the online figshare repository. Files on figshare will be made available approximately one week before the accompanying article so that the supplementary material can be attributed a unique DOI.

Online supplementary material will also carry the title and description provided during submission, so please ensure these are accurate and informative. Note that the Royal Society will not edit or typeset supplementary material and it will be hosted as provided. Please ensure that the supplementary material includes the paper details (authors, title, journal name, article DOI). Your article DOI will be 10.1098/rsob.2016[last 4 digits of e.g. 10.1098/rsob.20160049].

- 4) A media summary: a short non-technical summary (up to 100 words) of the key findings/importance of your manuscript. Please try to write in simple English, avoid jargon, explain the importance of the topic, outline the main implications and describe why this topic is newsworthy.

Images

Data-Sharing

It is a condition of publication that data supporting your paper are made available. Data should be made available either in the electronic supplementary material or through an appropriate

repository. Details of how to access data should be included in your paper. Please see <http://royalsocietypublishing.org/site/authors/policy.xhtml#question6> for more details.

Data accessibility section

Sincerely,

The Open Biology Team
<mailto:openbiology@royalsociety.org>

Reviewer(s)' Comments to Author:

Referee: 1

Comments to the Author(s)

Nicely written introduction very wide ranging and perhaps a bit too long. It emphasises how integrins are functionally important in many processes as they not only are agents to attach a cell to the ECM they are also involved in intercellular signalling, partially indirectly, by "promoting adhesion to the ECM so that cells are kept close to the source of signals". No surprise then that flies lacking integrins are devastated. So, the approach here, a familiar and well justified one, is to make small clones lacking integrin function *mys*⁻, so that late requirements can be studied in some isolation. The focus of attention is on ommatidial rotation, a late and interesting process where small groups of cells rotate en bloc. This local loss of *mys* affects rotation in the imaginal discs, while by the adult eye stage the *mys*⁻ clones have become a complete mess.

More gentle interference with the myospheroid gene (localised and late overexpression under sevenless control *sev*>*mys*) has a milder consequence causing defects of rotation and other problems with the adult eye pattern. These flies can be used as a sensitised background to test for the role of other proteins.

Evidence for link between integrins and ECM being required for rotation is reinforced when other proteins involved in attachment to the ECM are also removed – the relatively weak mutant phenotype induced by overexpression of *mys* is aggravated. The authors then build the evidence that attachment to the ECM is instrumental in rotation by removing these other proteins in clones and finding rotation defects. These methods demonstrate interactive requirement, they are not so good at working out what the integrins and the other genes do, particularly as the effect of overexpression is itself a mutant phenotype, and disturbing rotation is relatively unspecific. Nevertheless the work raises some interesting if only suggestive ideas.

The authors then turn their attention to PCP using the overexpression of *sev*>*mys* flies as a background. The key protein in PCP is Frizzled and its removal does not work on this assay. Surprising..... maybe the authors should emphasise in the discussion that this jigsaw puzzle piece does not fit. However, the loss of Vang, another PCP protein of less centrality than Fz, does

increase rotation defects. Even halving the amount of Flamingo has an effect in some special flies overexpressing *mys* and this adds to the argument of an involvement of PCP in the rotation process.

There are some relevant experiments that are missing, for example do *nemo* and *E-cad* mutations increase the phenotype of *sev*>*mys*? What about testing the interaction the other way around? – are PCP defects increased in a *myo/+* background? Do *myo/+* flies affect PCP non-autonomous signalling?

Overall the authors have done a careful investigation of genetic interactions to build an argument that some PCP proteins are involved, with the ECM, in rotation. The authors should make it clearer that the investigation uses a rather blunt scalpel: these are compromised eyes, the experiments combine one genetic sickness with another to deduce mutual dependency. Yes they use the conventional experimental language of developmental genetics but the arguments that they make are not watertight and some recognition of this should be made. Also, we are not much wiser about what the wildtype mechanism is. Nevertheless I think the results are useful, they widen the picture of rotation in an interesting way. Moreover, the whole paper is clear and well written, the technical quality of the work is high and well presented with beautiful images, detailed and convincing analysis of ommatidial orientations and we recommend that the paper should be accepted subject to some revision.

About Figure 5:

The authors claim that during a short period there is an accumulation of *myo* protein around R1 and R6. However, this is difficult to see in the figure. The authors should consider using a better image. Even if this finding is correct, the relevance is not clear, is the reader being taken on a rather foggy trip for no purpose?

Referee: 2

Comments to the Author(s)

Thuveson and colleagues report on the function of integrins and the ECM during *Drosophila* eye development, in particular during ommatidial rotation. They use loss of function clones and genetic interactions with overexpressed dominant negative forms of integrins to show that integrins are essential for proper ommatidial rotation and interact with planar cell polarity signaling components, which are known to be important for ommatidial rotation. They show that integrin localization changes dynamically during rotation and accumulates on cell membranes facing outside of ommatidial clusters.

This is a beautifully illustrated and well-presented paper describing specific *in vivo* functions for integrins and ECM components. The specific localization of integrins and how this is affected by Drok activation is particularly interesting and provides new insights into the difficult question of how ECM-integrins cause specific effects. I have only a minor suggestion: the rotation defects in integrin mutant clones is clear, however, the quantification as presented does not really illustrate the randomness and disorganization of the tissue. Maybe this could be illustrated by tabulating the difference in angle between neighboring clusters.

Referee: 3

Comments to the Author(s)

This paper explores the role of integrins in eye development, in particular its role in ommatidial rotation. The authors show that integrins are needed for the normal progression of ommatidial rotation, using a clones of mutant allele of *betaPS/mys*, and overexpression of *betaPS/mys*

constructs that cause dominant negative effects. The authors explore the pathway of this integrin function using genetic interactions, which show that reduction of a number of components of the integrin machinery, actin machinery and PCP machinery are able to enhance the dominant negative integrin construct phenotypes. This is a useful data set, but does not contain any surprises, and also falls short of providing a clear mechanism for how integrins contribute to ommatidial rotation. The value of the data set could be improved by 3 straightforward revisions.

1. The paper seems to be hinting that loss of integrin is unusual in that rotation is accelerated, whereas other mutations that impair rotation just reduce the rate. The authors need to be more explicit about this potential key difference and describe which mutants show accelerated rotation or retarded rotation, both in the published literature and the experiments presented in this paper. For example is there also acceleration in the perlecan and wing blister mutants shown in Fig. 6? Is this a unique feature of loss of integrin mediated adhesion? This might imply that rotation is primarily driven by lateral movement, and the basal attachment provides useful resistance to regulate the rate of movement. Is it possible to image the cell shape from apical to basal, and see whether the rotation is ahead at the top or bottom of the cell? This relates to recent work showing that convergent extension can be lead by junction exchange either apically or basally.

2. Expand Table 1 to include mutations that do not enhance the phenotype. Just showing mutations that enhance the phenotype leaves open the possibility than any mutation will do so. Showing the specificity of this enhancement is therefore valuable, and it is just as interesting to know the mutations that do not enhance as those that do enhance.

3. Allow a better comparison between the data in Table 1 and Fig 7.E. Using a different dominant negative construct for the genetic interaction experiments is a very useful addition, and the differences are especially interesting. However both data sets need to be displayed in the same way so we can directly compare them. Either both in a Figure or both in Table 1.

Minor errors

Intro.

p4 Cooper and Bray reference should be the 1999 one, not the 2000 one

p6, 2nd para. Beta-nu is not only functional in the midgut; there is evidence the beta-nu functions in haematopoietic cells, e.g Nagaosa et al., 2011 DOI10.1074/jbc.M110.204503

p13,14 the symbol for lamininA is LanA; lama is a different gene

Author's Response to Decision Letter for (RSOB-19-0148.R0)

See Appendix A.

Decision letter (RSOB-19-0148.R1)

23-Jul-2019

Dear Professor Mlodzik

We are pleased to inform you that your manuscript entitled "Integrins are required for synchronous ommatidial rotation in the *Drosophila* eye linking Planar Cell Polarity signaling to the extracellular matrix" has been accepted by the Editor for publication in Open Biology.

Sincerely,

The Open Biology Team
mailto: openbiology@royalsociety.org

Appendix A

Point-by-point response to reviewer comments (our responses in *blue italics*)

Rev 1:

Nicely written introduction very wide ranging and perhaps a bit too long. It emphasises how integrins are functionally important in many processes as they not only are agents to attach a cell to the ECM they are also involved in intercellular signalling, partially indirectly, by “promoting adhesion to the ECM so that cells are kept close to the source of signals”. No surprise then that flies lacking integrins are devastated. So, the approach here, a familiar and well justified one, is to make small clones lacking integrin function */mys-*, so that late requirements can be studied in some isolation. The focus of attention is on ommatidial rotation, a late and interesting process where small groups of cells rotate en bloc. This local loss of */mys* affects rotation in the imaginal discs, while by the adult eye stage the */mys-* clones have become a complete mess.

More gentle interference with the myospheroid gene (localised and late overexpression under *sevenless* control */sev>mys/*) has a milder consequence causing defects of rotation and other problems with the adult eye pattern. These flies can be used as a sensitised background to test for the role of other proteins.

Evidence for link between integrins and ECM being required for rotation is reinforced when other proteins involved in attachment to the ECM are also removed—the relatively weak mutant phenotype induced by overexpression of */mys/* is aggravated. The authors then build the evidence that attachment to the ECM is instrumental in rotation by removing these other proteins in clones and finding rotation defects. These methods demonstrate interactive requirement, they are not so good at working out what the integrins and the other genes do, particularly as the effect of overexpression is itself a mutant phenotype, and disturbing rotation is relatively unspecific. Nevertheless the work raises some interesting if only suggestive ideas.

The authors then turn their attention to PCP using the overexpression of */sev>mys/* flies as a background. The key protein in PCP is Frizzled and its removal does not work on this assay. Surprising..... maybe the authors should emphasise in the discussion that this jigsaw puzzle piece does not fit. However, the loss of Vang, another PCP protein of less centrality than Fz, does increase rotation defects. Even halving the amount of Flamingo has an effect in some special flies overexpressing */mys* and this adds to the argument of an involvement of PCP in the rotation process. *We thank the reviewer for their thorough and very positive evaluation of our manuscript. We are grateful for their constructive comments to help further improve our paper. Please see below for specific responses.*

There are some relevant experiments that are missing, for example do */nemo/* and E-cad mutations increase the phenotype of */sev>mys*? /What about testing the interaction the other way around? — are PCP defects increased in a */myo/+* background? Do */myo/+* flies affect PCP non-autonomous signalling?

We appreciate the suggestions of the reviewer. We have tested for interactions between sev-Mys and many genes, including the ones mentioned by the reviewer. These genes do not interact in the dosage sensitivity assay. We have added these (and several others) in Table 1 and in the text.

Overall the authors have done a careful investigation of genetic interactions to build an argument that some PCP proteins are involved, with the ECM, in rotation. The authors should make it clearer that the investigation uses a rather blunt scalpel: these are compromised eyes, the experiments combine one genetic sickness with another to deduce mutual dependency. Yes they use the conventional experimental language of developmental genetics but the arguments that they make are not

watertight and some recognition of this should be made. Also, we are not much wiser about what the wildtype mechanism is. Nevertheless I think the results are useful, they widen the picture of rotation in an interesting way. Moreover, the whole paper is clear and well written, the technical quality of the work is high and well presented with beautiful images, detailed and convincing analysis of ommatidial orientations and we recommend that the paper should be accepted subject to some revision.

We thank the reviewer for pointing out the strengths, and also highlighting potential inherent shortcomings due to the experimental approach.

We have addressed these in the discussion (pg 16/17) and trust that this adds to the clarity of the paper.

About Figure 5:

The authors claim that during a short period there is an accumulation of myo protein around R1 and R6. However, this is difficult to see in the figure. The authors should consider using a better image. Even if this finding is correct, the relevance is not clear, is the reader being taken on a rather foggy trip for no purpose?

We have moved the R1/R6 specific data from the main text (and Figure 5) to a new Supplemental Figure to avoid that it could detract the reader from the main message of the paper.

Rev 2:

Thuveson and colleagues report on the function of integrins and the ECM during *Drosophila* eye development, in particular during ommatidial rotation. They use loss of function clones and genetic interactions with overexpressed dominant negative forms of integrins to show that integrins are essential for proper ommatidial rotation and interact with planar cell polarity signaling components, which are known to be important for ommatidial rotation. They show that integrin localization changes dynamically during rotation and accumulates on cell membranes facing outside of ommatidial clusters.

This is a beautifully illustrated and well-presented paper describing specific in vivo functions for integrins and ECM components. The specific localization of integrins and how this is affected by Drok activation is particularly interesting and provides new insights into the difficult question of how ECM-integrins cause specific effects. I have only a minor suggestion: the rotation defects in integrin mutant clones is clear, however, the quantification as presented does not really illustrate the randomness and disorganization of the tissue. Maybe this could be illustrated by tabulating the difference in angle between neighboring clusters.

We thank the reviewer for their very positive assessment of our work and their praise of the quality of the data and the data presentation.

The suggestion on tabulating/presenting the angle differences between neighboring clusters is very useful. We have added this data to the revised figure 3, new panel Fig 3C.

Rev 3:

This paper explores the role of integrins in eye development, in particular its role in ommatidial rotation. The authors show that integrins are needed for the normal progression of ommatidial rotation, using a clones of mutant allele of betaPS/mys, and overexpression of betaPS/mys constructs that cause dominant negative effects. The authors explore the pathway of this integrin function using genetic interactions, which show that reduction of a number of components of the integrin machinery, actin machinery and PCP machinery are able to enhance the dominant negative integrin construct

phenotypes. This is a useful data set, but does not contain any surprises, and also falls short of providing a clear mechanism for how integrins contribute to ommatidial rotation. The value of the data set could be improved by 3 straightforward revisions.

We are grateful to the reviewer for their very positive assessment of our paper and their constructive comments on how to further improve it.

1. The paper seems to be hinting that loss of integrin is unusual in that rotation is accelerated, whereas other mutations that impair rotation just reduce the rate. The authors need to be more explicit about this potential key difference and describe which mutants show accelerated rotation or retarded rotation, both in the published literature and the experiments presented in this paper. For example is there also acceleration in the perlecan and wing blister mutants shown in Fig. 6? Is this a unique feature of loss of integrin mediated adhesion? This might imply that rotation is primarily driven by lateral movement, and the basal attachment provides useful resistance to regulate the rate of movement. Is it possible to image the cell shape from apical to basal, and see whether the rotation is ahead at the top or bottom of the cell? This relates to recent work showing that convergent extension can be lead by junction exchange either apically or basally.

We have added a paragraph in the discussion (pg 16/17) to address the differences and similarities between the Integrin and ECM rotation phenotypes with other rotation associated genes and defects.

2. Expand Table 1 to include mutations that do not enhance the phenotype. Just showing mutations that enhance the phenotype leaves open the possibility than any mutation will do so. Showing the specificity of this enhancement is therefore valuable, and it is just as interesting to know the mutations that do not enhance as those that do enhance.

We have added a number of genes that were tested and did not enhance the phenotype.

3. Allow a better comparison between the data in Table 1 and Fig 7.E. Using a different dominant negative construct for the genetic interaction experiments is a very useful addition, and the differences are especially interesting. However both data sets need to be displayed in the same way so we can directly compare them. Either both in a Figure or both in Table 1.

The data presented in Fig 7E has also been added to Table 1 to allow a more direct comparison.

Minor errors

Intro.

p4 Cooper and Bray reference should be the 1999 one, not the 2000 one

Thank you for pointing this out; this has been corrected.

p6, 2nd para. Beta-nu is not only functional in the midgut; there is evidence the beta-nu functions in haematopoietic cells, e.g Nagaosa et al., 2011 DOI10.1074/jbc.M110.204503

We have added the mention and reference to the function in hematopoietic cells.

p13,14 the symbol for lamininA is LanA; lama is a different gene

Thank you for pointing this out; this has been corrected.